# How Vietnamese Export Firms Faced Financial Distress during COVID-19? A Bayesian Small Sample Analysis

**Thanh Dan Bui [1] and Nguyen Ngoc Thach [2],***

1   Finance Faculty, Ho Chi Minh University of Banking, Ho Chi Minh City 70000, Vietnam
2   Asian Journal of Economics and Banking, Ho Chi Minh University of Banking,
    Ho Chi Minh City 70000, Vietnam
*   Correspondence: thachnn@hub.edu.vn

**Abstract:** A crucial role belongs to export firms in the export-led growth model of Vietnam. The COVID-19 disease has posed a serious challenge to the Vietnamese economy, having negatively impacted its influential export sector. However, investigating this export sector encounters small sample issues within the frequentist framework. So, by adopting a Bayesian approach, this study intends to explore the impact of COVID-19-specific factors on the financial distress of the export companies listed on the Vietnam stock exchange. In case frequentist and Bayesian estimation with non-informative priors cannot acquire statistical efficiency due to problems with insufficient data, thoughtful Bayesian analysis can yield meaningful outcomes. A Bayesian logistic regression model was employed to analyze the scarce data, and the posterior estimates of all the model parameters achieved significantly show that the main operating cash flow ratio to total debt, company size, and retained return on total assets are all negatively related to financial distress. Interestingly, financial leverage positively affects financial distress because a benefit from a tax shield could not compensate for the adverse impact of a debt burden, as suggested in the trade-off theory, which demonstrates the specific financial situation of the companies when the COVID-19 pandemic was in full swing.

**Keywords:** financial distress; export company; Bayesian estimation; small sample; informative priors; COVID-19

**JEL Classification:** C00; C11; C29

## 1. Introduction

Export activities of countries worldwide, including Vietnam, have been severely affected by the complicated developments of the COVID-19 disease. Vietnam has been implementing export-led economic growth, with its trade openness reaching 210% in 2021 (World Bank 2022). So, export firms play a central role in Vietnam's economic development. When a severe crisis took place for these firms, it adversely affected the whole economy. The policies of closing national borders and restricting the circulation of goods and services have enormously reduced trade activities. Since March 2020, the COVID-19 outbreak has disrupted the supply chain of goods and services in Vietnam. Major export markets of Vietnam have considerably decreased import demand. When national borders were declared closed to prevent the disease's spread, large orders were reported to be delayed. Vietnamese export firms faced many difficulties when they had to maintain production activities in the context of rapidly increasing production costs. The requirements of isolation and social distancing make it difficult for businesses to reach customers. More specifically, throughout the pandemic, many Vietnamese export businesses have encountered financial issues, which run a high risk of bankruptcy. The analysis of the financial distress of Vietnamese export companies is attracting much attention, but it is impossible to solve this problem in times of COVID-19 due to the limitation of data. To the best of our knowledge,

there have been no studies on financial hardship during the COVID-19 period for export companies. It is noted that most previous studies on corporate financial distress used descriptive analyses or frequentist methods with large datasets (e.g., Ernawati et al. 2018; Osama and Bassam 2019; Dirman 2020; Trung et al. 2022; and many others). The Bayesian approach is a visible alternative to frequentist methods in small-sample research. Hence, in this study, we choose the Bayesian logistic regression method to explore the impact of specific factors on the financial distress of the biggest export companies listed on the Vietnamese stock market. Our contributions are as follows: financial distress at the peak of the COVID-19 disaster in 2021, when the Delta coronavirus variant was spreading extremely fast, is first studied for export firms in an emerging economy; clearly, COVID-19 had specific effects on the financial position of export firms, and, second, in terms of methodology, the thoughtful Bayesian approach used in our study appears more appropriate than traditional frequentist methods, which, as a rule, require a large data sample with dummy variables included in a model to obtain significant estimates.

## 2. Literature Review

### 2.1. Trade-Off Theory

Kraus and Litzenberger (1973) developed the trade-off theory from the capital structure theory. According to the capital structure theory, when tax is included, there is a tax benefit from using debt, where the optimal capital structure is 100% debt (Modigliani and Miller 1958; Modigliani and Miller 1963). The trade-off theory accounts for financial distress by selecting an appropriate level of debt to achieve the largest firm value. This theory gives the maximum level of debt of a firm, exceeding which it may fall into financial distress. Thus, the trade-off theory is more applicable than the M&M theory. Besides, according to Andrade and Kaplan (1998), a corporation experiences financial crisis when it is unable to pay its debts. Failing to pay a debt and trying to restructure debt to avoid default are two examples of financial hardship. The first type of financial distress is when a corporation defaults on interest payments or violates debt covenants. Purnanandam (2008) goes on to show that attempts to restructure debt in order to prevent default are a trait of financial distress, which corresponds to the second category. Financial distress, as opposed to bankruptcy, is characterized by losses, subpar performance, and negative cumulative earnings for at least one or more straight years.

### 2.2. Empirical Research

There are a great number of empirical studies on corporate financial distress. However, most studies applied traditional frequentist methods with long time series data sets. For example, among the featured studies on financial distress, Tinoco and Wilson (2013), based on a sample of 23,218 year-round observations of listed companies for the period 1980–2011, using a logistic model to test the proposed hypotheses, discovered that operating cash flow on total debt negatively impacts to financial distress. Ninh et al. (2018) used a sample of 800 Vietnamese businesses from the years 2003 to 2016, with a total of 6736 observations, to investigate various models for default prediction. Four accounting proxies are shown by the empirical results of this study to have a negative correlation with default risk. According to these data, financial distress is positively correlated with leverage ratio and is less likely to occur in Vietnamese enterprises that have higher financial liquidity, asset productivity, solvency, and profitability. Large Vietnamese firms also have a lower default risk than smaller ones. Shrivastava et al. (2018) used a panel dataset extracted from Capital IQ to perform corporate distress prediction for an emerging economy such as India. EBITDA to tax ratio, working capital ratio, and size have a negative impact on distress, thereby lowering default probability, while profitability measures, viz., profit margin and retained earnings to assets ratios, the leverage ratio, and debt to asset ratio have the inverse impact on distress. Ernawati et al. (2018) used a logistic regression model to analyze the impact of financial ratios and corporate governance on the financial difficulties of 310 non-financial companies listed on the Indonesia Stock Exchange (IDX) in December 2016.

The results exhibited that the ratios of short-term debt to total assets, total liabilities to total assets, and market book value are positively related to financial distress, whereas, by contrast, sales to total assets, earnings before interest, and taxes to total assets have the opposite effect. Vo et al. (2019) considered the financial distress of approximately 800 listed firms at the industry level in Vietnam over the last decade. Two periods are considered, namely during the Global Financial Crisis (GFC) (2007–2009) and post-GFC (2010–2017). The logit regression technique is used to estimate alternative models based on accounting and market factors,. In conclusion, the earnings before interest and taxes over the total asset (EBIT/TA) variable have a negative impact on financial distress. Osama and Bassam (2019) examined the determinants of financial distress in MENA countries. By using an annual dataset of 796 listed MENA companies between 2009 and 2017, the results showed that working capital/total assets; income before interest and taxes/total liabilities; retained earnings/total assets; revenue/total assets; operating cash flow/revenue; and operating cash flow/total assets have adverse effects on the financial distress of firms. Dirman (2020) provides an insightful understanding of the effects of profitability, liquidity, leverage, company size, and natural cash flow on financial distress using the multiple linear regression method with a data sample of 90 manufacturing companies in the primary and chemical industries listed on the Indonesian Stock Exchange (IDX) for 2016–2018. The research results showed that profitability positively impacts financial distress; liquidity, leverage, and free cash flow do not affect financial distress; firm size harms financial distress. Trung et al. (2022) determined the factors influencing the financial distress of 623 Vietnamese firms during the period 2014–2019 by using a Bayesian binary logistic regression model and found that financial distress increases as financial leverage rises, while firm size, net working capital to current assets, retained earnings to total assets and earnings before interest, and taxes to total assets has a negative impact. Truong (2022) used data from 240 Vietnamese listed non-financial enterprises from 2010 to 2019 to show how corporate governance affects the relationship between macro and micro causes producing financial hardship. An endogenous switching regression model is used in the study to examine the marginal benefits of various corporate governance methods (ESRM). According to the results, return on asset, firm size, and the ratio of the gross profit to the total revenue have a negative correlation with financial distress; meanwhile, the percentage of the book value of total debt on the book value of the total assets and the ratio of current assets to current liabilities have the opposite impact.

The abovementioned investigations addressed corporate financial distress utilizing mainly frequency-based approaches with adequate sample sizes, especially since the research period was often before the COVID-19 outbreak. Because of a limited research population, small sample issues usually occur in the applications of frequentist methods. With small sample sizes, power issues make it impossible to obtain significant results using frequentist estimators. Several econometricians, such as Lee and Song (2004), Van de Schoot et al. (2015), Zondervan-Zwijnenburg et al. (2017), and Smid et al. (2019) have emphasized power failure within the frequentist framework. Small sample frequentist analyses often result in "non-convergence, inadmissible parameter solutions, and inaccurate estimates" (Smid et al. 2019), "low statistical power and level of relative bias" (Van de Schoot et al. 2015; Zondervan-Zwijnenburg et al. 2017). In terms of structural and variance components, popular frequentist regressions such as least square (LS), maximum likelihood (ML), or restricted maximum likelihood (REML) underperform on Bayesian estimators using non-informative, informative (specific), and data-driven priors, respectively, in small sample research (Bradley 1978); In particular, Van de Schoot et al. (2015), Miočević et al. (2017), and Zondervan-Zwijnenburg et al. (2017) stressed that informative priors improve power for structural parameters in Bayesian inference compared to frequentist inference. Remarkably, naïve Bayesian estimation produces even more biased results than frequentist methods (Depaoli and Clifton 2015; McNeish 2016; Holtmann et al. 2016). Herein, the fundamental reason behind significant bias levels is that posteriors are dominated by priors in small sample research (Lee and Song 2004; McNeish 2016). Indeed, non-informative prior settings

yield a wide range of plausible parameter values. Most significantly, Bayesian estimation with informative priors outperforms naïve Bayesian and frequentist estimation in most simulations' investigations (Depaoli and Clifton 2015; Van de Schoot et al. 2015; McNeish 2016; Miočević et al. 2017).

Hence, the current research conducts a Bayesian logistic regression model specifying informative Cauchy priors on a small sample of Vietnam's 34 biggest export companies in 2021. This study differs from previous works in the following points: (i) our work is the first to explore specific factors, particularly financial leverage, determining the financial distress of export firms in the context of an emerging Asian economy when the COVID-19 pandemic reached its peak in the Southeast Asian region; specifically, under the conditions of international border closures and rapid coronavirus spread, export firms were faced with huge financial hardship; (ii) By implementing a Bayesian MCMC simulation study based on an informative prior specification, we generalize a vital conclusion that, contrary to frequentist inference, Bayesian inference with informative priors can provide meaningful results, irrespective of data sample size, namely, with a small cross-section sample of 2021 for export firms in Vietnam, a Bayesian logistic regression model can provide meaningful outcomes.

## 3. Model Specification and Data Sample

### 3.1. Bayesian Markov Chain Simulations

Since the end of the twentieth century, a new paradigm has been observed in statistical analysis with an increase in the methodological and empirical application of the Bayesian approach (e.g., Nguyen et al. 2019; Kreinovich et al. 2019; Thach et al. 2021; Thach et al. 2022). Such progress took place owing to the recent advancements in software programs and the advantageous properties of the Bayesian paradigm compared with traditional frequentist settings. The usefulness of Bayesian analysis is, first, to estimate complex models or models too demanding for frequentist estimators; second, Bayesian estimates are straightforwardly interpreted; third, a simple probability rule, called Bayes' rule, applies to all parametric regression models; fourth, flexibility involving model uncertainty in Bayesian analysis (Van de Schoot et al. 2015). More crucially, the thoughtful Bayesian approach allows for a solution to several statistical issues appearing in frequentist inference due to data limitation, such as multicollinearity, endogeneity, or type I and II errors, when common tests in IV regression perform poorly in small samples. Bayesian estimation using well-specified priors can give significant results in case data are scarce. Many simulation studies highlighted a marked preference for informative over non-informative default priors (e.g., Depaoli and Clifton 2015; McNeish 2016; Smid et al. 2019).

A common problem in Bayesian simulation studies is the prior choice of parameters in the model. Conservative researchers tend to choose empirical (data-driven) or non-informative priors, which is contrasted with the Bayesian framework. Certainly, non-informative, default prior settings cannot solve some common regression issues, particularly the separation problem in logistic regression. However, as background knowledge is lacking, no general guidelines for eliciting informative priors are available. In this investigation, some recommendations from Gelman et al. (2008) are utilized to fairly assign informative Cauchy priors to the parameters of our logistic model. The authors propose the Cauchy distribution with a center of 0 and a scale of 2.5, which in the simplest framework is a longer-tailed version of the distribution attained by assuming one-half additional success and one-half additional failure in a logistic regression. With cross-validation on a corpus of datasets, the Cauchy class of prior distributions is shown to outperform different Gaussian and Laplace priors. The advantage of the Cauchy class of priors is always to provide solutions, even in the case that complete separation occurs in logistic regression (a common issue, even in the case of a large sample size and a small number of regressors).

### 3.2. Model and Data

Mimicking previous studies (Osama and Bassam 2019; Ernawati et al. 2018; Dirman 2020; Trung et al. 2022; Tinoco and Wilson 2013), the authors specify an econometric model as follows:

$$FDIS_{it} = \alpha + \beta_1 OCF_{it} + \beta_2 LEV_{it} + \beta_3 SIZE_{it} + \beta_4 NWCA_{it} + \beta_5 RETA_{it} + \beta_6 OCFTL_{it} + u_{it} \qquad (1)$$

where $FDIS_{it}$ is the likelihood of financial distress in company i in 2021; $OCF_{it}$ is the operating cash flow of company i in 2021; $LEV_{it}$ is the financial leverage of company i in 2021; $SIZE_{it}$ is the size of company i in 2021; $NWCA_{it}$ is the ratio of net working capital to short-term assets of company i in 2021; $RETA_{it}$ is the ratio of retained earnings to total assets of company i in 2021; $OCFTL_{it}$ is the ratio of operating cash flow to total debt of company i in 2021.

Firstly, a standard logistic regression model is estimated. Furthermore, we run a Bayesian logistic regression model using default normal priors. By default, normal priors with a mean of 0 and a variance of 10,000 are specified for the intercept and regression coefficients, respectively. Finally, a thoughtful Bayesian model with informative prior distributions is specified. Gelman et al. (2008) suggested weakly informative Cauchy priors for the regression coefficients in case standardized data are used so that all continuous variables have a mean of 0 and a standard deviation of 0.5. In particular, a scale of 10 for the intercept and a scale of 2.5 for the regression coefficients are chosen. This choice comes from the observation that within the unit change of each predictor, a response change of 5 units on the logistic scale will move the response probability from 0.01 to 0.5 and from 0.5 to 0.99. Since the covariates are centered at 0, the Cauchy priors are centered at 0 too.

We employ a cross-sectional dataset of 34 Vietnamese listed export companies, with the time being 2021, when the Delta variant of the coronavirus dramatically affected Vietnam, especially its export sector. The data are collected from the financial statements of 33 export companies listed on the Vietnamese stock market. As recommended by Lee and Song (2004), with a ratio of observations and parameters lower than 5:1, frequentist methods cannot yield meaningful results. In our case, this ratio is 34:8, less than 5:1. Hence, a Bayesian approach needs to be assessed.

We standardize the covariates of the model to specify a common prior to the parameters, as proposed by Gelman et al. (2008) (Table 1). Raftery (1996) favored this viewpoint as well. After being standardized, all the non-binary variables have a mean of 0 and a standard deviation of 0.5. For robustness purposes, we withhold the first and last observations from being used in the simulation.

**Table 1.** Standardized data.

| Variable | Obs | Mean | Std. Dev. | Min | Max |
|:---:|:---:|:---:|:---:|:---:|:---:|
| OCF | 33 | −1.9893 | 0.5 | −1.196 | 1.214 |
| LEV | 33 | −14.004 | 0.5 | −1.281 | 0.735 |
| SIZE | 33 | 2.707 | 0.5 | −0.662 | 1.292 |
| NWCA | 33 | −34.152 | 0.5 | −0.767 | 1.341 |
| RETA | 33 | −9.960 | 0.5 | −0.622 | 1.494 |
| OCFT | 33 | −40.535 | 0.5 | −0.950 | 2.521 |

Source: Calculated by the authors.

### 3.3. Variables and Hypotheses

Response variable: In this study, the likelihood of financial distress (FDIS) is represented by the company's earnings after tax (EAT) and undivided profit (UP). The FDIS variable takes a value of 1 if a company does not encounter financial difficulty (EAT and UP > 0) and 0 if the company is faced with a financial problem (EAT and UP < 0)

(Table 2). Most of the analyzed export companies managed their business well in 2021. Only 3 companies have a negative net profit after tax and retained earnings.

**Table 2.** Description of variables and research hypotheses.

| Variable | Description | Measure | Hypothesis | Source |
|----------|-------------|---------|:----------:|--------|
| | | **Dependent variable** | | |
| FDIS | The likelihood of financial distress | Coded: 1 if EAT and UP > 0 0 if EAT and UP < 0 | | Osama and Bassam (2019), Ernawati et al. (2018), Dirman (2020), Trung et al. (2022), Tinoco and Wilson (2013) |
| | | Independent variables | | |
| OCF | Operating cash flow | $OCF = \frac{\text{Operating cash flow}}{\text{Total Asset}}$ | - | Osama and Bassam (2019) |
| LEV | Financial leverage | $LEV = \frac{\text{Total debt}}{\text{Total Asset}}$ | + | Ernawati et al. (2018) |
| SIZE | Company size | SIZE = Logarithm (Total Asset) | - | Dirman (2020) |
| NWCA | Net working capital to current assets | $NWCA = \frac{\text{Net working capital}}{\text{Current Assets}}$ | - | Trung et al. (2022) |
| RETA | Retained return on total asset | $RETA = \frac{\text{Retained return}}{\text{Total Asset}}$ | - | Osama and Bassam (2019), Trung et al. (2022) |
| OCFTL | Operating cash flow to total debt ratio | $OCFTL = \frac{\text{Operating cash flow}}{\text{Total debt}}$ | - | Tinoco and Wilson (2013) |

Source: Compiled by the authors.

Predictor variables: In the context of COVID-19, we focus on predictors related to the organization's internal resources because internal elements significantly help businesses manage cash flow and other financial issues.

Operating cash flow to total assets (OCF): The OCF variable is measured by operating cash flow to total assets, showing cash outflows compared to cash inflows from operating activities during a year. A higher ratio of operating cash flow to total assets indicates that a company's business operations are efficient so that it can avoid a financial crisis (Osama and Bassam 2019).

**Hypothesis H1.** *Operating cash flow to total assets is negatively correlated to financial distress.*

Financial leverage (LEV): Variable LEV is proxied by total debt to total assets. According to the trade-off theory, as a company increases its financial leverage, its financial obligations rise, which decreases its financial safety. If a company ineffectively uses its debt, then the rise of debt would increase financial distress (Ernawati et al. 2018). According to Truong (2022), companies with strong or weak corporate governance experience lower profits when taking on significant debt. A debt load brought on by high debt raises the possibility of default and financial trouble.

**Hypothesis H2.** *Financial leverage is positively related to financial distress.*

Company size (SIZE): The SIZE variable is a log of total assets. A small business finds it difficult to obtain outside funding owing to its smaller asset base than large corporates (Dirman 2020). Compared to large businesses, small ones are more susceptible to financial problems since they can persuade investors less.

**Hypothesis H3.** *Firm size negatively contributes to financial distress.*

Net working capital (NWCA): VLDR is proxied by net working capital to short-term assets. This index was suggested by Trung et al. (2022) to assess the factors contributing to financial distress. Insolvency is a state of financial difficulty. The authors view insolvency as a signal similar to firm liquidity (Graham et al. 2011).

**Hypothesis H4.** *The ratio of net working capital to short-term assets is negatively related to financial distress.*

Retained return on total assets (RETA): The RETA variable is measured by retained earnings to total assets. Osama and Bassam (2019) argued that businesses that cannot fund their ideas by relying on internally generated capital due to low retained earnings on total assets must borrow money and accrue interest. The increased debt eventually results in higher financing costs, which raises the likelihood of default.

**Hypothesis H5.** *The ratio of retained earnings to total assets is negatively correlated to financial distress.*

Operating cash flow to total debt (OCFTL): The OCFTL variable is presented by the operating cash flow ratio to total debt. This ratio demonstrates a company's capacity to pay down its obligations out of operating cash flow (Tinoco and Wilson 2013).

**Hypothesis H6.** *The ratio of operating cash flow to total debt negatively contributes to financial distress.*

## 4. Results and Discussion

### 4.1. Bayesian Simulation Results

The separation problem appears when we implement frequentist estimation. That is because some observations are completely determined as the continuous covariates have many repeating values. More specifically, due to insufficient sample data in the current study, the frequentist logistic regression yields non-significant estimates for all the variables in the model.

Furthermore, the MCMC simulation results from the naïve Bayesian model (that is, with default prior settings) are reported in Table 3.

**Table 3.** Naïve Bayesian estimation.

| Variable | Reg. Coeff. | SD | Posterior SE | Credibility Interval |
|---|---|---|---|---|
| FDIS | | | | |
| OCF | 13.731 | 39.688 | 3.649 | −62.164, 94.868 |
| LEV | −87.132 | 52.478 | 4.631 | −203.645, 3.518 |
| SIZE | 116.369 | 47.884 | 5.669 | 26.688, 212.532 |
| NWCA | 34.705 | 52.136 | 2.925 | −139.192, 64.617 |
| RETA | 109.862 | 47.250 | 6.077 | 33.401, 211.857 |
| OCFT | −103.218 | 57.886 | 7.109 | −224.660, 0.498 |
| Intercept | 112.259 | 33.622 | 4.692 | 53.090, 182.096 |

Note: Credibility interval is the 95% probability that a parameter lies between two values in the population; SD is the standard deviation; SE is the standard error. Source: Calculated by the authors.

Finally, the MCMC simulation results of the thoughtful Bayesian model with fairly informative Cauchy priors are shown in Table 4.

**Table 4.** Thoughtful Bayesian estimation with informative priors.

| Variable | Reg. Coeff. | SD | Posterior SE | Probability of the Effect Being More than Zero | Credibility Interval |
|---|---|---|---|---|---|
| FDIS | | | | | |
| OCF | 0.500 | 2.574 | 0.012 | 58% | −4.259, 6.121 |
| LEV | −3.816 | 5.217 | 0.034 | 87% * | −16.709, 1.650 |
| SIZE | 8.116 | 11.309 | 0.115 | 90% | −1.497, 38.363 |
| NWCA | 0.074 | 3.180 | 0.015 | 49% * | −6.820, 5.962 |
| RETA | 12.402 | 11.469 | 0.127 | 98% | 0.385, 41.850 |
| OCFT | −5.607 | 6.809 | 0.038 | 93% * | −23.883, 0.846 |
| Intercept | 10.534 | 7.291 | 0.098 | | 3.194, 29.200 |

Note: * indicates a probability of the effect being less than zero. Source: Calculated by the authors.

Remarkably, in contrast to frequentist estimation, the Bayesian model with the informative prior settings (Table 4) can provide significant estimates for all the model parameters. Furthermore, the log marginal–likelihood estimate of the Bayesian model using informative priors is higher than that of the naïve Bayesian model (−5.4 versus −8.4). So, the former outperforms the latter (Table 3). To use the simulation results of the Bayesian model with informative prior settings, we must check MCMC convergence. Graphical tests via trace and autocorrelation plots are performed (Figure 1a,b). Then, we will check MCMC convergence based on a popular formal test (Table 5. None of the MCMC convergence diagnostics revealed any anomalies.

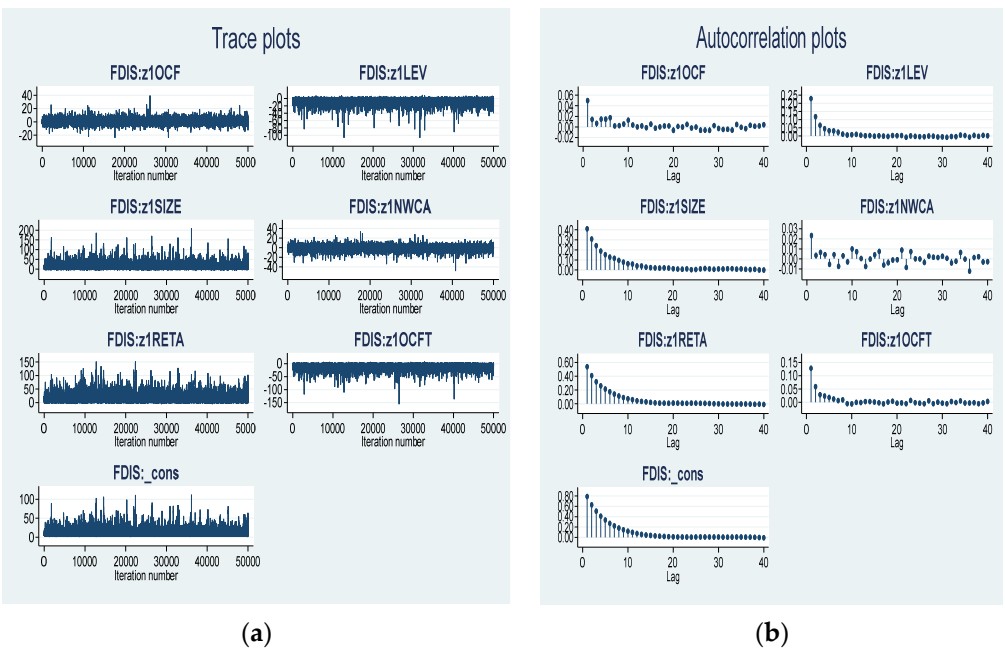

(**a**)                                    (**b**)

**Figure 1.** (**a**) Trace plots and (**b**) autocorrelation plots.

**Table 5.** Effective sample size.

| FDIS | ESS | Corr. Time | Efficiency |
|---|---|---|---|
| OCF | 44,324.63 | 1.13 | 0.887 |
| LEV | 23,688.35 | 2.11 | 0.474 |
| SIZE | 9763.64 | 5.12 | 0.195 |
| NWCA | 47,774.44 | 1.05 | 0.956 |
| RETA | 8219.32 | 6.08 | 0.164 |
| OCFT | 32,447.96 | 1.54 | 0.649 |
| Intercept | 5530.50 | 9.04 | 0.111 |

Source: Calculated by the authors.

*4.2. Interpreting Results*

Compared with the naïve Bayesian and frequentist estimates, Bayesian logistic estimation using informative Cauchy priors could provide significant conclusions in the context of a small sample size. According to the MCMC simulation outcomes shown in Table 4, the authors discover a negative association between operating cash flow to total assets ratio and financial distress, which is consistent with hypothesis H1. That is because business segments were disrupted in 2021 when export companies faced problems caused by COVID-19, such as social distance, labor shortage, scarce raw materials, etc. The government issued Directive No. 11/CT-TTg (Government 2020) on urgent tasks to overcome difficulties in the COVID-19 epidemic. Furthermore, the State Bank of Vietnam directed banks to meet capital needs for business. Therefore, export companies could continue their business activities. Notably, the financial leverage ratio is positively and strongly (with a probability of 87%) correlated with financial distress. This result is in agreement with hypothesis H2. If firms wanted to expand activities or hold business, they had to take more borrowings. As the trade-off theory indicates, by using debt, a firm can gain a tax shield from interest; however, if using too much debt, it would face financial distress. In agreement with hypothesis H3, firm size has a negative and very strong correlation (with a 90% probability) with financial distress. In 2021, the crisis challenged export companies when they acted with increased costs, and only big firms had more ability to overcome financial difficulties. Consistent with hypothesis H4, the ratio of net working capital to short-term assets (NWCA) is negatively correlated with financial distress. As figures show, exporting companies had very low net working capital under uncertainty during COVID-19. Low net working capital demonstrates that regular capital is exactly enough to cover a company's current assets. Consequently, the company is under pressure to turn around short-term loans and find alternative sources of capital, falling into financial distress. The ratio of retained earnings to total assets exerts a negative and very strong (with a probability of 98%) effect on financial distress, which is consistent with hypothesis H5. A probable reason is that during the pandemic, efficient business operations minimize the possibility of financial distress for a company. In contrast to hypothesis H6, the study found a positive and strong relationship (with a probability of 93%) between the ratio of operating cash flow to total debt and financial distress. During the pandemic, managers, to cover the costs of maintaining a company, had to sell off the assets. So, companies with a low cash flow ratio from operating activities must borrow money from outside, quickly falling into financial difficulties.

**5. Conclusions and Policy Implications**

*5.1. Conclusions*

By implementing a Bayesian MCMC simulation analysis of the impacts of COVID-19-specific factors on the financial distress of 34 export companies listed on the stock market of Vietnam during COVID-19, the current research shows that in the context of a small sample dataset, specifying informative Cauchy priors, Bayesian MCMC simulations perform better

than both naïve Bayesian and frequentist estimation. More importantly, the Bayesian outcomes are significant for all the parameters in the model. In the simulation study, all the specific factors included in the analysis affect the probability of financial distress of the listed export companies. The Bayesian outcomes demonstrate that, except for the operating cash flow ratio to total liabilities and financial leverage, operating cash flow, company size, retained profit ratio to total assets, and net working capital ratio to current assets are negatively correlated with financial distress. The authors suggest some recommendations for managers to regularize the capital structure of their companies to reduce the probability of financial distress in the context of an economic crisis. The novelty of this study is that for export companies, the negative effect of a debt burden is more than a benefit from a tax shield in COVID-19 times.

*5.2. Policy Implications*

Below are recommendations provided in light of the current export development in Vietnam after COVID-19:

Operating cash flow: A company's business strategy needs to be built on a scientific foundation, showing high flexibility to cope with the export industry's stagnation resulting from the economy's recession.

Financial leverage: Businesses must strictly control their payments to creditors to increase company value through financial leverage. They must analyze the connection between asset profitability and debt costs to determine whether the debt is effectively used and to maintain the financial conditions required by lending institutions.

Company size: To reduce the likelihood of financial trouble, a company must actively accumulate experience in both risk and overall financial management.

Net working capital: A company should design an effective investment strategy. For this, it needs to consider money mobilization, current financial capacity, and lack of capital and analyze the availability of capital sources.

Retained earnings: A company should use retained earnings to enhance its size and develop a strategy for effectively using the extra funds with a higher return for the shareholders. As a result, exporting businesses need to have a roadmap for raising cash at each development stage; they should avoid making unreasonably large investments.

Operating cash flow on total liabilities: To assure their ability to earn cash and balance payment risk and capital efficiency when increasing external funding, businesses concentrate on the balance between receipts and expenditures, as well as effective management of working capital.

**Author Contributions:** Conceptualization; investigation; resources; data curation; writing—original draft preparation; writing—review and editing: T.D.B.; methodology; software; validation; formal analysis; visualization; supervision; project administration; funding acquisition: N.N.T. All authors have read and agreed to the published version of the manuscript.

**Funding:** This study receives no external funding.

**Institutional Review Board Statement:** Not applicable.

**Informed Consent Statement:** Not applicable.

**Data Availability Statement:** The data that support the findings of this study are available from the corresponding author upon reasonable request.

**Conflicts of Interest:** The authors declare no conflict of interest.

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
