# Peer review of "How Vietnamese Export Firms Faced Financial Distress during COVID-19? A Bayesian Small Sample Analysis"

_economies, doi:10.3390/economies11020041_

Round 1

Author Response

Dear respected reviewer,

Thanks a lot for your valuable comments. We revised our manuscript accordingly.

Reviewer 2 Report

The paper is not adequately developed and still needs much work to be up to its promise and clear. The paper has major issues that negatively affect its incremental contribution to the literature review

-          The introduction is missing number of important elements. For example, what is the motivation of investigating the export firms? Why Vietnam ?

-          Authors should discuss their contribution in the introduction.

-          The paper lacks a rigorous contribution in its present form. For instants . how the study is different from the below references:

Osama, E. A., & Bassam, L. (2019). Predicting Financial Distress For Listed MENA Firms. International Journal of Accounting and 369 Financial Reporting, 2(9), 51-75

Tinoco, M.H., Wilson, N. (2013). Financial Distress and Bankruptcy Prediction Among Listed Companies Using Accounting, Market 381 And Macroeconomic Variables. International Review of Financial Analysis, No. 30, pp. 394-419

Trung, N.D., Thanh, B.D., Phuong, B.N.M., Lan, L.T. (2022). Factors Influencing The Financial Distress Probability Of Vietnam 383 Enterprises. In International Econometric Conference of Vietnam, Springer, Cham, pp. 635-649.

-          The analysis is very basic , the authors could have tried harder by adopting a range of additional methods that used in previous literature.

-          The paper is missing a persuasive hypothesis development. Why do some firms-specific factors influence the financial disaster ?

I am concern with endogeneity issue in this paper. For example the financial disaster could be the cause of deciding to retain return. There is no evidence analysis that has been provided to solve this problem

Author Response

(The authors gave the same response as above.)

Round 2

Author Response

Response to review (round 2)

 - Since, according to the authors, the study of Vietnam is one of the contributions of the study, I think this should be reflected in the title itself;

Response:

We add “Vietnamese” in the paper title:

How Vietnamese Export Firms Faced Financial Distress during Covid-19? A Bayesian Small Sample Analysis

 - Regarding the contribution, it seems to me that greater reflection is needed. Why is studying a small sample with these particularities relevant, for whom, and to what extent is this study and these specific conclusions relevant, and in what way? What is gained by analyzing a 1-year sample of 34 companies in which only 3 are in default?

Response:

We study COVID-19 impacts on export firms at the peak of the pandemic in Vietnam in 2021, in the context of the delta coronavirus variant’s rapid spread. So, we should use a cross-section of 2021, but not a panel dataset with a dummy variable as frequentist researchers often do, facing a small sample. For this purpose, a Bayesian approach is more appropriate. The study has both methodological and policy implications:

Methodologically: The study is addressed Bayesians and frequentists to enrich their methodological tools. It also provided policy implications for export and other firms in VietNam and other developing countries when facing such a crisis as COVID-19.

Policy implications: Out of 34 firms in the sample, 3 is in default, but many others were on the verge of bankruptcy under the influence of the delta variant. So, policy implications are for both healthy and “sick” firms.

 -Concerning the models that, despite being frequent, the overwhelming number of articles on non-compliance refer to them, does it make sense not to mention them?

Indeed, there is a rich body of literatute on this topic. We consider all these empirical studies as evidence that there are many authors interested in the topic of financial distress but most previous studies used frequentist methods based on large data samples. When researchers study a crisis, they should use a dummy variable and regularities derived is affected by the entire data distribution. But few studies used Bayesian methods. Therefore, our research paper with Bayesian method will be the point of difference in this topic. In this study,  we review only key studies, especially those, from which we select appropriate variables for our Bayesian model.

  • One of the aspects that seems to me to still need improvement are the differences compared to other studies, it is mentioned that it studies specific aspects, in particular financial leverage, is this true?

Response:

First, ours differ from the previous studies in methodology, that is, we perform a Bayesian simulation study, which is more suitable for a small sample analysis.

Second, we mention that this study looks at specific aspects. This is shown through the independent variables, especially their calculation. All are specific factors show the internal financial position of the company. Regarding financial leverage, this variable is negatively correlated to financial distress in the normal economy as firms used a tax shield. However, in a crisis, firms, which take much borrowings, more often face debt burden and so this variable has the opposite effect.

- It seems to me that there should be greater clarification and support for the default criterion adopted, is it a sufficiently rigorous criterion to classify a company in financial difficulties?

Response:

We understand that, during the difficult economic period caused by the epidemic, if a company  maintain their business operations, have positive profit after tax and retained earnings, then it is not facing financial difficulties and vice versa. This is a COVID19-specific criterion.

- As for the explanatory variables, it is still not clear why these variables are used and not others, does it make sense to use the OCF and OCFTL simultaneously? Why the acronym VLDR?

Response:

All are specific factors included in the our model show the internal financial position of the company. The simultaneous use of OCF and OCFTL will further exploit the financial aspects of the company as they are different measures; OCF rises but OCFTL can decrease due to a sharp increase in debt. In addition, we find the abbreviation "VLDR" pure Vietnamese. So we changed the abbreviation to "NWCA''.

- In terms of conclusions, the justification for the impact of the OCFTL variable, taking into account the impact of the OCF variable, does not seem clear to me.

Response:

OCF measures operating cash flow to total assets, while OCFTL measures operating cash flow to total debt. Therefore, the effects of these two variables differ.

We thank reviewers and editors for your valuable suggestions and comments.

Response to review (round 2)

 - Since, according to the authors, the study of Vietnam is one of the contributions of the study, I think this should be reflected in the title itself;

Response:

We add “Vietnamese” in the paper title:

How Vietnamese Export Firms Faced Financial Distress during Covid-19? A Bayesian Small Sample Analysis

 - Regarding the contribution, it seems to me that greater reflection is needed. Why is studying a small sample with these particularities relevant, for whom, and to what extent is this study and these specific conclusions relevant, and in what way? What is gained by analyzing a 1-year sample of 34 companies in which only 3 are in default?

Response:

We study COVID-19 impacts on export firms at the peak of the pandemic in Vietnam in 2021, in the context of the delta coronavirus variant’s rapid spread. So, we should use a cross-section of 2021, but not a panel dataset with a dummy variable as frequentist researchers often do, facing a small sample. For this purpose, a Bayesian approach is more appropriate. The study has both methodological and policy implications:

Methodologically: The study is addressed Bayesians and frequentists to enrich their methodological tools. It also provided policy implications for export and other firms in VietNam and other developing countries when facing such a crisis as COVID-19.

Policy implications: Out of 34 firms in the sample, 3 is in default, but many others were on the verge of bankruptcy under the influence of the delta variant. So, policy implications are for both healthy and “sick” firms.

 -Concerning the models that, despite being frequent, the overwhelming number of articles on non-compliance refer to them, does it make sense not to mention them?

Indeed, there is a rich body of literatute on this topic. We consider all these empirical studies as evidence that there are many authors interested in the topic of financial distress but most previous studies used frequentist methods based on large data samples. When researchers study a crisis, they should use a dummy variable and regularities derived is affected by the entire data distribution. But few studies used Bayesian methods. Therefore, our research paper with Bayesian method will be the point of difference in this topic. In this study,  we review only key studies, especially those, from which we select appropriate variables for our Bayesian model.

  • One of the aspects that seems to me to still need improvement are the differences compared to other studies, it is mentioned that it studies specific aspects, in particular financial leverage, is this true?

Response:

First, ours differ from the previous studies in methodology, that is, we perform a Bayesian simulation study, which is more suitable for a small sample analysis.

Second, we mention that this study looks at specific aspects. This is shown through the independent variables, especially their calculation. All are specific factors show the internal financial position of the company. Regarding financial leverage, this variable is negatively correlated to financial distress in the normal economy as firms used a tax shield. However, in a crisis, firms, which take much borrowings, more often face debt burden and so this variable has the opposite effect.

- It seems to me that there should be greater clarification and support for the default criterion adopted, is it a sufficiently rigorous criterion to classify a company in financial difficulties?

Response:

We understand that, during the difficult economic period caused by the epidemic, if a company  maintain their business operations, have positive profit after tax and retained earnings, then it is not facing financial difficulties and vice versa. This is a COVID19-specific criterion.

- As for the explanatory variables, it is still not clear why these variables are used and not others, does it make sense to use the OCF and OCFTL simultaneously? Why the acronym VLDR?

Response:

All are specific factors included in the our model show the internal financial position of the company. The simultaneous use of OCF and OCFTL will further exploit the financial aspects of the company as they are different measures; OCF rises but OCFTL can decrease due to a sharp increase in debt. In addition, we find the abbreviation "VLDR" pure Vietnamese. So we changed the abbreviation to "NWCA''.

- In terms of conclusions, the justification for the impact of the OCFTL variable, taking into account the impact of the OCF variable, does not seem clear to me.

Response:

OCF measures operating cash flow to total assets, while OCFTL measures operating cash flow to total debt. Therefore, the effects of these two variables differ.

We thank reviewers and editors for your valuable suggestions and comments.

Response to review (round 2)

 - Since, according to the authors, the study of Vietnam is one of the contributions of the study, I think this should be reflected in the title itself;

Response:

We add “Vietnamese” in the paper title:

How Vietnamese Export Firms Faced Financial Distress during Covid-19? A Bayesian Small Sample Analysis

 - Regarding the contribution, it seems to me that greater reflection is needed. Why is studying a small sample with these particularities relevant, for whom, and to what extent is this study and these specific conclusions relevant, and in what way? What is gained by analyzing a 1-year sample of 34 companies in which only 3 are in default?

Response:

We study COVID-19 impacts on export firms at the peak of the pandemic in Vietnam in 2021, in the context of the delta coronavirus variant’s rapid spread. So, we should use a cross-section of 2021, but not a panel dataset with a dummy variable as frequentist researchers often do, facing a small sample. For this purpose, a Bayesian approach is more appropriate. The study has both methodological and policy implications:

Methodologically: The study is addressed Bayesians and frequentists to enrich their methodological tools. It also provided policy implications for export and other firms in VietNam and other developing countries when facing such a crisis as COVID-19.

Policy implications: Out of 34 firms in the sample, 3 is in default, but many others were on the verge of bankruptcy under the influence of the delta variant. So, policy implications are for both healthy and “sick” firms.

 -Concerning the models that, despite being frequent, the overwhelming number of articles on non-compliance refer to them, does it make sense not to mention them?

Indeed, there is a rich body of literatute on this topic. We consider all these empirical studies as evidence that there are many authors interested in the topic of financial distress but most previous studies used frequentist methods based on large data samples. When researchers study a crisis, they should use a dummy variable and regularities derived is affected by the entire data distribution. But few studies used Bayesian methods. Therefore, our research paper with Bayesian method will be the point of difference in this topic. In this study,  we review only key studies, especially those, from which we select appropriate variables for our Bayesian model.

  • One of the aspects that seems to me to still need improvement are the differences compared to other studies, it is mentioned that it studies specific aspects, in particular financial leverage, is this true?

Response:

First, ours differ from the previous studies in methodology, that is, we perform a Bayesian simulation study, which is more suitable for a small sample analysis.

Second, we mention that this study looks at specific aspects. This is shown through the independent variables, especially their calculation. All are specific factors show the internal financial position of the company. Regarding financial leverage, this variable is negatively correlated to financial distress in the normal economy as firms used a tax shield. However, in a crisis, firms, which take much borrowings, more often face debt burden and so this variable has the opposite effect.

- It seems to me that there should be greater clarification and support for the default criterion adopted, is it a sufficiently rigorous criterion to classify a company in financial difficulties?

Response:

We understand that, during the difficult economic period caused by the epidemic, if a company  maintain their business operations, have positive profit after tax and retained earnings, then it is not facing financial difficulties and vice versa. This is a COVID19-specific criterion.

- As for the explanatory variables, it is still not clear why these variables are used and not others, does it make sense to use the OCF and OCFTL simultaneously? Why the acronym VLDR?

Response:

All are specific factors included in the our model show the internal financial position of the company. The simultaneous use of OCF and OCFTL will further exploit the financial aspects of the company as they are different measures; OCF rises but OCFTL can decrease due to a sharp increase in debt. In addition, we find the abbreviation "VLDR" pure Vietnamese. So we changed the abbreviation to "NWCA''.

- In terms of conclusions, the justification for the impact of the OCFTL variable, taking into account the impact of the OCF variable, does not seem clear to me.

Response:

OCF measures operating cash flow to total assets, while OCFTL measures operating cash flow to total debt. Therefore, the effects of these two variables differ.

We thank reviewers and editors for your valuable suggestions and comments.

Response to review (round 2)

 - Since, according to the authors, the study of Vietnam is one of the contributions of the study, I think this should be reflected in the title itself;

Response:

We add “Vietnamese” in the paper title:

How Vietnamese Export Firms Faced Financial Distress during Covid-19? A Bayesian Small Sample Analysis

 - Regarding the contribution, it seems to me that greater reflection is needed. Why is studying a small sample with these particularities relevant, for whom, and to what extent is this study and these specific conclusions relevant, and in what way? What is gained by analyzing a 1-year sample of 34 companies in which only 3 are in default?

Response:

We study COVID-19 impacts on export firms at the peak of the pandemic in Vietnam in 2021, in the context of the delta coronavirus variant’s rapid spread. So, we should use a cross-section of 2021, but not a panel dataset with a dummy variable as frequentist researchers often do, facing a small sample. For this purpose, a Bayesian approach is more appropriate. The study has both methodological and policy implications:

Methodologically: The study is addressed Bayesians and frequentists to enrich their methodological tools. It also provided policy implications for export and other firms in VietNam and other developing countries when facing such a crisis as COVID-19.

Policy implications: Out of 34 firms in the sample, 3 is in default, but many others were on the verge of bankruptcy under the influence of the delta variant. So, policy implications are for both healthy and “sick” firms.

 -Concerning the models that, despite being frequent, the overwhelming number of articles on non-compliance refer to them, does it make sense not to mention them?

Indeed, there is a rich body of literatute on this topic. We consider all these empirical studies as evidence that there are many authors interested in the topic of financial distress but most previous studies used frequentist methods based on large data samples. When researchers study a crisis, they should use a dummy variable and regularities derived is affected by the entire data distribution. But few studies used Bayesian methods. Therefore, our research paper with Bayesian method will be the point of difference in this topic. In this study,  we review only key studies, especially those, from which we select appropriate variables for our Bayesian model.

  • One of the aspects that seems to me to still need improvement are the differences compared to other studies, it is mentioned that it studies specific aspects, in particular financial leverage, is this true?

Response:

First, ours differ from the previous studies in methodology, that is, we perform a Bayesian simulation study, which is more suitable for a small sample analysis.

Second, we mention that this study looks at specific aspects. This is shown through the independent variables, especially their calculation. All are specific factors show the internal financial position of the company. Regarding financial leverage, this variable is negatively correlated to financial distress in the normal economy as firms used a tax shield. However, in a crisis, firms, which take much borrowings, more often face debt burden and so this variable has the opposite effect.

- It seems to me that there should be greater clarification and support for the default criterion adopted, is it a sufficiently rigorous criterion to classify a company in financial difficulties?

Response:

We understand that, during the difficult economic period caused by the epidemic, if a company  maintain their business operations, have positive profit after tax and retained earnings, then it is not facing financial difficulties and vice versa. This is a COVID19-specific criterion.

- As for the explanatory variables, it is still not clear why these variables are used and not others, does it make sense to use the OCF and OCFTL simultaneously? Why the acronym VLDR?

Response:

All are specific factors included in the our model show the internal financial position of the company. The simultaneous use of OCF and OCFTL will further exploit the financial aspects of the company as they are different measures; OCF rises but OCFTL can decrease due to a sharp increase in debt. In addition, we find the abbreviation "VLDR" pure Vietnamese. So we changed the abbreviation to "NWCA''.

- In terms of conclusions, the justification for the impact of the OCFTL variable, taking into account the impact of the OCF variable, does not seem clear to me.

Response:

OCF measures operating cash flow to total assets, while OCFTL measures operating cash flow to total debt. Therefore, the effects of these two variables differ.

We thank reviewers and editors for your valuable suggestions and comments.

Response to review (round 2)

 - Since, according to the authors, the study of Vietnam is one of the contributions of the study, I think this should be reflected in the title itself;

Response:

We add “Vietnamese” in the paper title:

How Vietnamese Export Firms Faced Financial Distress during Covid-19? A Bayesian Small Sample Analysis

 - Regarding the contribution, it seems to me that greater reflection is needed. Why is studying a small sample with these particularities relevant, for whom, and to what extent is this study and these specific conclusions relevant, and in what way? What is gained by analyzing a 1-year sample of 34 companies in which only 3 are in default?

Response:

We study COVID-19 impacts on export firms at the peak of the pandemic in Vietnam in 2021, in the context of the delta coronavirus variant’s rapid spread. So, we should use a cross-section of 2021, but not a panel dataset with a dummy variable as frequentist researchers often do, facing a small sample. For this purpose, a Bayesian approach is more appropriate. The study has both methodological and policy implications:

Methodologically: The study is addressed Bayesians and frequentists to enrich their methodological tools. It also provided policy implications for export and other firms in VietNam and other developing countries when facing such a crisis as COVID-19.

Policy implications: Out of 34 firms in the sample, 3 is in default, but many others were on the verge of bankruptcy under the influence of the delta variant. So, policy implications are for both healthy and “sick” firms.

 -Concerning the models that, despite being frequent, the overwhelming number of articles on non-compliance refer to them, does it make sense not to mention them?

Indeed, there is a rich body of literatute on this topic. We consider all these empirical studies as evidence that there are many authors interested in the topic of financial distress but most previous studies used frequentist methods based on large data samples. When researchers study a crisis, they should use a dummy variable and regularities derived is affected by the entire data distribution. But few studies used Bayesian methods. Therefore, our research paper with Bayesian method will be the point of difference in this topic. In this study,  we review only key studies, especially those, from which we select appropriate variables for our Bayesian model.

  • One of the aspects that seems to me to still need improvement are the differences compared to other studies, it is mentioned that it studies specific aspects, in particular financial leverage, is this true?

Response:

First, ours differ from the previous studies in methodology, that is, we perform a Bayesian simulation study, which is more suitable for a small sample analysis.

Second, we mention that this study looks at specific aspects. This is shown through the independent variables, especially their calculation. All are specific factors show the internal financial position of the company. Regarding financial leverage, this variable is negatively correlated to financial distress in the normal economy as firms used a tax shield. However, in a crisis, firms, which take much borrowings, more often face debt burden and so this variable has the opposite effect.

- It seems to me that there should be greater clarification and support for the default criterion adopted, is it a sufficiently rigorous criterion to classify a company in financial difficulties?

Response:

We understand that, during the difficult economic period caused by the epidemic, if a company  maintain their business operations, have positive profit after tax and retained earnings, then it is not facing financial difficulties and vice versa. This is a COVID19-specific criterion.

- As for the explanatory variables, it is still not clear why these variables are used and not others, does it make sense to use the OCF and OCFTL simultaneously? Why the acronym VLDR?

Response:

All are specific factors included in the our model show the internal financial position of the company. The simultaneous use of OCF and OCFTL will further exploit the financial aspects of the company as they are different measures; OCF rises but OCFTL can decrease due to a sharp increase in debt. In addition, we find the abbreviation "VLDR" pure Vietnamese. So we changed the abbreviation to "NWCA''.

- In terms of conclusions, the justification for the impact of the OCFTL variable, taking into account the impact of the OCF variable, does not seem clear to me.

Response:

OCF measures operating cash flow to total assets, while OCFTL measures operating cash flow to total debt. Therefore, the effects of these two variables differ.

We thank reviewers and editors for your valuable suggestions and comments.

Response to review (round 2)

 - Since, according to the authors, the study of Vietnam is one of the contributions of the study, I think this should be reflected in the title itself;

Response:

We add “Vietnamese” in the paper title:

How Vietnamese Export Firms Faced Financial Distress during Covid-19? A Bayesian Small Sample Analysis

 - Regarding the contribution, it seems to me that greater reflection is needed. Why is studying a small sample with these particularities relevant, for whom, and to what extent is this study and these specific conclusions relevant, and in what way? What is gained by analyzing a 1-year sample of 34 companies in which only 3 are in default?

Response:

We study COVID-19 impacts on export firms at the peak of the pandemic in Vietnam in 2021, in the context of the delta coronavirus variant’s rapid spread. So, we should use a cross-section of 2021, but not a panel dataset with a dummy variable as frequentist researchers often do, facing a small sample. For this purpose, a Bayesian approach is more appropriate. The study has both methodological and policy implications:

Methodologically: The study is addressed Bayesians and frequentists to enrich their methodological tools. It also provided policy implications for export and other firms in VietNam and other developing countries when facing such a crisis as COVID-19.

Policy implications: Out of 34 firms in the sample, 3 is in default, but many others were on the verge of bankruptcy under the influence of the delta variant. So, policy implications are for both healthy and “sick” firms.

 -Concerning the models that, despite being frequent, the overwhelming number of articles on non-compliance refer to them, does it make sense not to mention them?

Indeed, there is a rich body of literatute on this topic. We consider all these empirical studies as evidence that there are many authors interested in the topic of financial distress but most previous studies used frequentist methods based on large data samples. When researchers study a crisis, they should use a dummy variable and regularities derived is affected by the entire data distribution. But few studies used Bayesian methods. Therefore, our research paper with Bayesian method will be the point of difference in this topic. In this study,  we review only key studies, especially those, from which we select appropriate variables for our Bayesian model.

  • One of the aspects that seems to me to still need improvement are the differences compared to other studies, it is mentioned that it studies specific aspects, in particular financial leverage, is this true?

Response:

First, ours differ from the previous studies in methodology, that is, we perform a Bayesian simulation study, which is more suitable for a small sample analysis.

Second, we mention that this study looks at specific aspects. This is shown through the independent variables, especially their calculation. All are specific factors show the internal financial position of the company. Regarding financial leverage, this variable is negatively correlated to financial distress in the normal economy as firms used a tax shield. However, in a crisis, firms, which take much borrowings, more often face debt burden and so this variable has the opposite effect.

- It seems to me that there should be greater clarification and support for the default criterion adopted, is it a sufficiently rigorous criterion to classify a company in financial difficulties?

Response:

We understand that, during the difficult economic period caused by the epidemic, if a company  maintain their business operations, have positive profit after tax and retained earnings, then it is not facing financial difficulties and vice versa. This is a COVID19-specific criterion.

- As for the explanatory variables, it is still not clear why these variables are used and not others, does it make sense to use the OCF and OCFTL simultaneously? Why the acronym VLDR?

Response:

All are specific factors included in the our model show the internal financial position of the company. The simultaneous use of OCF and OCFTL will further exploit the financial aspects of the company as they are different measures; OCF rises but OCFTL can decrease due to a sharp increase in debt. In addition, we find the abbreviation "VLDR" pure Vietnamese. So we changed the abbreviation to "NWCA''.

- In terms of conclusions, the justification for the impact of the OCFTL variable, taking into account the impact of the OCF variable, does not seem clear to me.

Response:

OCF measures operating cash flow to total assets, while OCFTL measures operating cash flow to total debt. Therefore, the effects of these two variables differ.

We thank reviewers and editors for your valuable suggestions and comments.

Response to review (round 2)

 - Since, according to the authors, the study of Vietnam is one of the contributions of the study, I think this should be reflected in the title itself;

Response:

We add “Vietnamese” in the paper title:

How Vietnamese Export Firms Faced Financial Distress during Covid-19? A Bayesian Small Sample Analysis

 - Regarding the contribution, it seems to me that greater reflection is needed. Why is studying a small sample with these particularities relevant, for whom, and to what extent is this study and these specific conclusions relevant, and in what way? What is gained by analyzing a 1-year sample of 34 companies in which only 3 are in default?

Response:

We study COVID-19 impacts on export firms at the peak of the pandemic in Vietnam in 2021, in the context of the delta coronavirus variant’s rapid spread. So, we should use a cross-section of 2021, but not a panel dataset with a dummy variable as frequentist researchers often do, facing a small sample. For this purpose, a Bayesian approach is more appropriate. The study has both methodological and policy implications:

Methodologically: The study is addressed Bayesians and frequentists to enrich their methodological tools. It also provided policy implications for export and other firms in VietNam and other developing countries when facing such a crisis as COVID-19.

Policy implications: Out of 34 firms in the sample, 3 is in default, but many others were on the verge of bankruptcy under the influence of the delta variant. So, policy implications are for both healthy and “sick” firms.

 -Concerning the models that, despite being frequent, the overwhelming number of articles on non-compliance refer to them, does it make sense not to mention them?

Indeed, there is a rich body of literatute on this topic. We consider all these empirical studies as evidence that there are many authors interested in the topic of financial distress but most previous studies used frequentist methods based on large data samples. When researchers study a crisis, they should use a dummy variable and regularities derived is affected by the entire data distribution. But few studies used Bayesian methods. Therefore, our research paper with Bayesian method will be the point of difference in this topic. In this study,  we review only key studies, especially those, from which we select appropriate variables for our Bayesian model.

  • One of the aspects that seems to me to still need improvement are the differences compared to other studies, it is mentioned that it studies specific aspects, in particular financial leverage, is this true?

Response:

First, ours differ from the previous studies in methodology, that is, we perform a Bayesian simulation study, which is more suitable for a small sample analysis.

Second, we mention that this study looks at specific aspects. This is shown through the independent variables, especially their calculation. All are specific factors show the internal financial position of the company. Regarding financial leverage, this variable is negatively correlated to financial distress in the normal economy as firms used a tax shield. However, in a crisis, firms, which take much borrowings, more often face debt burden and so this variable has the opposite effect.

- It seems to me that there should be greater clarification and support for the default criterion adopted, is it a sufficiently rigorous criterion to classify a company in financial difficulties?

Response:

We understand that, during the difficult economic period caused by the epidemic, if a company  maintain their business operations, have positive profit after tax and retained earnings, then it is not facing financial difficulties and vice versa. This is a COVID19-specific criterion.

- As for the explanatory variables, it is still not clear why these variables are used and not others, does it make sense to use the OCF and OCFTL simultaneously? Why the acronym VLDR?

Response:

All are specific factors included in the our model show the internal financial position of the company. The simultaneous use of OCF and OCFTL will further exploit the financial aspects of the company as they are different measures; OCF rises but OCFTL can decrease due to a sharp increase in debt. In addition, we find the abbreviation "VLDR" pure Vietnamese. So we changed the abbreviation to "NWCA''.

- In terms of conclusions, the justification for the impact of the OCFTL variable, taking into account the impact of the OCF variable, does not seem clear to me.

Response:

OCF measures operating cash flow to total assets, while OCFTL measures operating cash flow to total debt. Therefore, the effects of these two variables differ.

We thank reviewers and editors for your valuable suggestions and comments.

Response to review (round 2)

 - Since, according to the authors, the study of Vietnam is one of the contributions of the study, I think this should be reflected in the title itself;

Response:

We add “Vietnamese” in the paper title:

How Vietnamese Export Firms Faced Financial Distress during Covid-19? A Bayesian Small Sample Analysis

 - Regarding the contribution, it seems to me that greater reflection is needed. Why is studying a small sample with these particularities relevant, for whom, and to what extent is this study and these specific conclusions relevant, and in what way? What is gained by analyzing a 1-year sample of 34 companies in which only 3 are in default?

Response:

We study COVID-19 impacts on export firms at the peak of the pandemic in Vietnam in 2021, in the context of the delta coronavirus variant’s rapid spread. So, we should use a cross-section of 2021, but not a panel dataset with a dummy variable as frequentist researchers often do, facing a small sample. For this purpose, a Bayesian approach is more appropriate. The study has both methodological and policy implications:

Methodologically: The study is addressed Bayesians and frequentists to enrich their methodological tools. It also provided policy implications for export and other firms in VietNam and other developing countries when facing such a crisis as COVID-19.

Policy implications: Out of 34 firms in the sample, 3 is in default, but many others were on the verge of bankruptcy under the influence of the delta variant. So, policy implications are for both healthy and “sick” firms.

 -Concerning the models that, despite being frequent, the overwhelming number of articles on non-compliance refer to them, does it make sense not to mention them?

Indeed, there is a rich body of literature on this topic. We consider all these empirical studies as evidence that many authors are interested in financial distress, but most previous studies used frequentist methods based on large data samples. When researchers study a crisis, they should use a dummy variable, and the regularities derived are affected by the entire data distribution. But few studies used Bayesian methods. Therefore, our research paper with the Bayesian method will be the point of difference in this topic. In this study,  we review only key studies, especially those from which we select appropriate variables for our Bayesian model.

  • One of the aspects that seems to me to still need improvement are the differences compared to other studies, it is mentioned that it studies specific aspects, in particular financial leverage, is this true?

Response:

First, ours differ from the previous studies in methodology; that is, we perform a Bayesian simulation study, which is more suitable for small sample analysis.

Second, we mention that this study looks at specific aspects. This is shown through the independent variables, especially their calculation. All are specific factors that show the internal financial position of the company. Regarding financial leverage, this variable is negatively correlated to financial distress in the normal economy as firms use a tax shield. However, in a crisis, firms that take much borrowing more often face a debt burden, so this variable has the opposite effect.

- It seems to me that there should be greater clarification and support for the default criterion adopted, is it a sufficiently rigorous criterion to classify a company in financial difficulties?

Response:

We understand that, during the difficult economic period caused by the epidemic, if a company maintains its business operations and has positive profit after tax and retained earnings, it is not facing financial difficulties and vice versa. This is a COVID-19-specific criterion.

- As for the explanatory variables, it is still not clear why these variables are used and not others, does it make sense to use the OCF and OCFTL simultaneously? Why the acronym VLDR?

Response:

All are specific factors included in the our model show the internal financial position of the company. The simultaneous use of OCF and OCFTL will further exploit the financial aspects of the company as they are different measures; OCF rises, but OCFTL can decrease due to a sharp increase in debt. In addition, we find the abbreviation "VLDR" in pure Vietnamese. So we changed the abbreviation to "NWCA''.

- In terms of conclusions, the justification for the impact of the OCFTL variable, taking into account the impact of the OCF variable, does not seem clear to me.

Response:

OCF measures operating cash flow to total assets, while OCFTL measures operating cash flow to total debt. Therefore, the effects of these two variables differ.

We thank the reviewers and editors for your valuable suggestions and comments.

v
